# Are Current Protection Methods Ensuring the Safe Emancipation of Young Black Storks? Telemetry Study of Space Use by Black Storks (*Ciconia nigra*) in the Early Post-Breeding Period

**DOI:** 10.3390/ani14111558

**Published:** 2024-05-24

**Authors:** Dariusz Anderwald, Marek Sławski, Tomasz Zadworny, Grzegorz Zawadzki

**Affiliations:** 1Forest Experimental Station in Rogów, Akademicka 20, 95-063 Rogów, Poland; dariusz_anderwald@sggw.edu.pl; 2Institute of Forest Sciences, Warsaw University of Life Sciences, Nowoursynowska 159, 02-776 Warszawa, Poland; grzegorz_zawadzki@sggw.edu.pl; 3Regional Directorate for Environmental Protection in Łódź, ul. Traugutta 25, 90-113 Łódź, Poland; tomasz.zadworny@lodz.rdos.gov.pl

**Keywords:** nesting area, nesting phenology, zonal protection, space use, GPS-GSM logger

## Abstract

**Simple Summary:**

The black stork is a large woodland bird. There has been a steady decline in its numbers in Europe in recent years. The safe development and independence of young individuals in the post-breeding period are crucial for the conservation of this species’ population. This research uses GPS loggers to track young storks’ use of space from the time of their first flight attempts until they finally leave the nest. During this period, the birds spend most of their time in an area less than 200 m away from the nest. They often rest outside the nest, with three-quarters of their resting places being within 500 m of the nest. Ensuring that the birds have peace and quiet during the nesting season in a zone 500 m from the nest is a good conservation strategy for this rare species.

**Abstract:**

The black stork is a protected species in Poland, and its numbers have declined significantly in recent years. The protection of nesting sites during the period of growth and independence of young birds is crucial for the population. In 2022–2023, 34 young storks were equipped with GPS-GSM backpack loggers. On average, birds had left the nest by the 87th day of life. In the period between the first flight attempt and the final abandonment of the nest, the birds spent 82% of their time in a zone up to 200 m from the nest. During the period of independence, resting areas played an important spatial role, 75% of which were located within 500 m of the nest. As the young birds grew older, their area of activity gradually increased. Differences in nesting phenology were observed depending on the geographical location of the nest. A shorter migration route from the wintering grounds allowed for earlier breeding. As a result, the young birds begin to fledge earlier. The data collected confirm the validity of designating protective zones with 500 m radii around nests and the need to maintain them from the beginning of the breeding season in March until the end of August.

## 1. Introduction

The rapid development of telemetry, resulting in smaller transmitter sizes and lower costs, in recent years has allowed its wider use in the study of the biology of mainly large animal species, such as large raptors [1,2,3,4,5] and storks (*Ciconia* sp.) [6,7,8]. Moreover, technological advances are enabling the use of telemetry transmitters for new types of research. One species that has received much attention in studies using GPS transmitters is the black stork (BS), *Ciconia nigra* [7,8,9,10]. Studies using GPS-GSM loggers on BS have mainly focused on autumn and spring migrations and their long-distance movements from breeding sites to African wintering grounds [6,11,12,13].

The BS is protected in the European Union under the Birds Directive [14]. In Poland, it is a rare species under strict protection, and its breeding sites are also protected. The range of the BS extends almost the whole Palearctic (from the Iberian Peninsula up to Eastern China and the Korean Peninsula) and part of South Africa [15,16], with the population estimated to be 9800–13,900 pairs across the European range [17]. The BS population in Poland is estimated to be 1300–1900 breeding pairs [18]. However, a decline in the population has been observed in Poland over the last decade, possibly as high as 30% [19]. Due to the negative population trends of the BS, various conservation programs dedicated to this species are being implemented [20,21]. In many countries, the BS is under strict species protection [22]. Passive protection is contained in the Birds Directive, including protection of the BS’s breeding habitat and restrictions on the intensity of forest management. Active protection activities are also being undertaken. Artificial nests are being built, and existing nests are being improved to ensure nesting safety [20,23,24]. To improve foraging conditions, several projects have made efforts to raise water levels in watercourses, create small floodplains, and deforest marshy areas [21,25]. The recommendations for foresters’ activities, such as prescribing the preservation of potential nest trees, are being modified. Another group of activities is related to the increasing number of studies of the BS, mostly based on tracking the routes of their autumn and spring migrations [10,26,27]. In Poland, protective buffer zones are designated around BS’s nests. Forestry and other economic activities within 200 m of BS nests are prohibited all year round. During the breeding season (15 March–31 August), this buffer zone is increased, and the area 500 m around the nest is protected from human activities. Such provisions of the Law on Nature Protection are intended to ensure tranquility for this rare species and allow successful breeding [22]. This legal status has been in effect for about 30 years and was introduced based on the best information from ornithologists at the time. However, we know that the number of storks in Poland is declining [18]. We therefore thought it was worth investigating whether this was due to poor protection of the nests of this skittish bird. This study asks the following questions: are the established extent and timing of protective restrictions optimal and can they guarantee these birds the right conditions for reproduction? Does such programmed protection provide the right conditions for the development of young birds? Finally, how does the process of becoming independent work?

The low abundance of BSs, as well as the declines reported in Eastern European countries [28,29,30,31,32,33,34], make it necessary to better understand the detailed aspects of breeding biology that influence the health of BS populations. Recently published research results indicate the low survival rates of young birds [27,31], making it even more important to study the effectiveness of conservation methods. Loggers are the best tool for collecting data about the lives of young BSs during the first period after they gain the ability to fly, with a low impact on the behavior studied.

Besides some papers describing the movements of young BSs during autumn migration [27,32], information on space use by BSs during the breeding season is scarce [34]. Mobility during the initial period of independence of young BSs at the end of the breeding season until the day they definitively leave the nest and begin their migration has not yet been analyzed. This process is relatively well understood in raptors, in which several phases preceding dispersal have been identified [5,35]. One of these is a long period of flocking to parental territories under the care of adult birds, where the young are taught to forage for food. A characteristic of the BS is that the young stay in or near their nests, where they are fed until the day they take up migration, most often undertaken alone [36]. Studies conducted on White Storks (*Ciconia ciconia*) highlight the importance of the short pre- and post-fledging periods in determining the subsequent survival of juvenile birds in the wild [37]. Rotics et al. [37] have also indicated the unique merits of using tracking data to relate early-life development and behavior to the fate of an individual.

In the present study, we investigated the spatial behavior of young BSs in the stands surrounding their nests during the post-nesting period, from their first flight until they left the nest. We focused on this key period of the first flights of young birds because it determines the conditions of the birds before migration. We were particularly interested in the following issues:

(1) Verifying the validity of designating a 200 m, year-round protection zone and a periodic 500 m zone;

(2) Describing the process of independence of young black storks from the moment of achieving the ability to fly progresses;

(3) Validating the timing of the periodic 500 m buffer zone around the nest (15 March–31 August).

## 2. Materials and Methods

### 2.1. Photo Dataset

We carried out the research in the forests of western and central-eastern Poland. From 2022 to 2023, camera traps were placed at 40 black stork nests with high breeding statuses. We used SF 3.5CG camera traps (SiFar SY Electronic Technology, Shenzhen, China). The monitored nests were those where the Forest Service had previously found chicks or the presence of a stork pair. The devices used were small and had a protective camouflage casing. They were equipped with an infrared backlight with a ‘no glow LED’ function, making them invisible to the birds. Wireless data transmission over a distance was facilitated by an integrated GPS/GPRS module using LTE/GSM technology, commonly used in mobile telephony, to transmit text and photo messages. We set the traps between 1 and 15 March, before the birds arrived, and removed them after the birds had departed, after 15 September. The camera traps were positioned at a distance of 1.5 to 2.5 m from the nests, at an angle that allowed observation of the inside of the nest. The camera traps collected a single photo each day. The use of Multimedia Messaging Service to transmit photos allowed for the ongoing analysis of the nestlings’ presence, age, and condition, which helped in selecting an appropriate date for the installation of loggers. We did not analyze any other data from the camera traps.

### 2.2. GPS Telemetry Dataset

In 2022–2023, 34 large BS chicks aged 45–55 days were equipped with GPS-GSM backpack loggers. In the end, 18 nests were deemed suitable for putting transmitters on the birds (Figure 1). Fourteen nests were placed on oaks (*Quercus* spp.) and four were built on Scots pines (*Pinus sylvestris*). Each record included the date and time of observation, geographic coordinates, altitude, and flight speed. The devices were primarily set up in the western part of Poland in the first year and in central and southeastern Poland in the second year. Teflon ribbons were used to attach loggers to the backs of young storks, similar to the ‘Y’ method recommended for white-tailed eagles by Buehler et al. [38]. This method ensured that the solar panel was well exposed to light, regardless of the chick’s position in the nest or the time of ringing or mounting the devices. The backpack loggers operated continuously and had sufficient voltage, even in heavily shaded areas such as under the treetop canopy where nests were typically located. Data were collected at intervals of 15 to 60 min, depending on the availability of light and the panel’s charge. All specimens were equipped with DRUID GPS loggers, mainly FLEX 2G NG (Druid Technology, Shenzhen, China) models weighing between 23 g and 24 g.

### 2.3. Data Analysis

Telemetry analyses were conducted using data collected from the movements of 30 black storks tracked between 2022 and 2023. The data cover the life span of young BSs from the 65th day of the chicks’ life to the day they definitively left the nest. None of the young BSs started flying before the 65th day of life. We considered the first flight to be a movement of a bird over 50 m from the nest. The age of the chicks was determined based on the development of their plumage and beak and wing length measurements collected during ringing and transmitter placement. The hatching date of young BSs was reconstructed within an accuracy of 2 days based on the age of the chicks at the time of ringing [39]. These data were then used to calculate the age of the birds on the day they left the nest and were subsequently used in statistical analyses. The day of hatching and the day of departure were converted to the next day of the year and used in this form in the statistical analyses. The day on which the birds left the nest and did not return for the night was considered as the date of departure from the nest. Movement analyses excluded four young birds whose flushing was recorded by a camera at the nest due to human interference, causing them to leave the nest before reaching 70 days of age.

GPS logger data were collected in .csv and .kml formats in the WGS84 dataset. QGIS version 3.32.3 Lima software was used for analysis and visualization. When analyzing the presence of BSs in the stands, records indicating the movement of individuals at a speed greater than or equal to 5 km/h were discarded. As a result, only stands where young storks actually stayed for longer periods and did not just fly past at random were analyzed. The geographical coordinates of the recording made it possible to assign the bird’s position to a specific stand according to the Forest Data Bank. We used a time- and frequency-based analysis of records in annular buffers around the nest, as opposed to the commonly used methods based on kernels and MPC. This approach allowed us to directly relate the results to zonal protection of BSs based on circular zones of 200 and 500 m radii. The following distance buffers around the known position of the nest tree were determined as 0–50 m, >50–100 m, >100–200 m, >200–300 m, >300–500 m, and >500–1000 m; the presence of birds within each of these distances from the nest was analyzed. For some analyses, the original data set was aggregated into wider annuli around the nest to better match the protective buffer zones. When calculating the time spent by an individual in the distance intervals (ring buffers) from the nests, the time difference between the end point and the starting point of each individual’s presence in a given distance interval from the nest was calculated, and the time of these visits was summed for all designated distance intervals. The time between records recorded in different zones was ignored, and records that were only recorded during a given visit in a given buffer were discarded. To find the most important places where the studied BSs spent most of their time away from the nest, only records where the birds moved at a speed of less than 5 km/h were used. Clustering algorithms available in QGIS were then used twice to search for groups of recorded points with respect to mutual distance and recording time. The first time, ST-DBSCAN clustering was used with the minimum cluster size set to 3 objects, the maximum distance between cluster points set to 50 m, and the maximum time difference between points set to 4 h. Clustering was performed again with the same parameters, but the algorithm omitted the time data. To obtain objects with polygonal geometry, a convex hull was created from the groups of points obtained. Areas where birds were frequently observed outside the nest that were identified in this way were called nodal points. The polygons assigned to the nodal points were combined with the descriptions of the forest from the Forest Data Bank [40] to characterize the stands containing the nodal points. The age and dominant tree species of these stands were determined.

### 2.4. Statistical Analysis

A chi-square test was used to analyze the BSs’ preference for using the area surrounding the nest. The percentage of the area of each buffer (data aggregated into the following annuli: 0–100, >100–200, >200–300, >300–500, >500–1000 m) around the nest was compared to the area of a circle with a radius of 1000 m (3.14 km^2^), as well as the percentage of time that individuals spent in these buffers. The Kruskal–Wallis test was used to determine whether there were differences between nests in terms of the young birds’ lengths of flights, frequency of buffer use, and the age at which they left the nest, depending on the number of young storks in the brood. Post hoc pairwise comparisons were performed using Dunn’s test with Bonferroni correction to analyze statistically significant differences. Spearman’s correlation coefficient was used to evaluate the relationships between latitude and longitude and the calendar day of chick hatching, the calendar day of young’s flight from the nest, the age at which young storks left the nest and the day of BS life with average daily flight length and daily total flights. We used Spearman rank correlation due to the large number of outlier observations and non-normal data distribution. A probabilistic logistic regression model was used to estimate the probability of young storks leaving the nest as a function of age and to calculate the cut-off date when the chance of the young flying out exceeded 50%. Logistic regression was also used to calculate the probability of the presence of young storks as a function of the distance from the nest. Statistical analyses were performed in R and RStudio in packages *dunn.test*, *sjplot*, and *Tidyverse*, and visualization was performed using the *ggpubr* and *ggplot2* packages [41].

## 3. Results

### 3.1. Use of Space

Thirty-four transmitters were placed on birds from 18 nests. The size of the analyzed broods ranged from two to four young. Between one and four loggers were placed on chicks in individual nests. Young birds started to make short flights near the nest starting from 65 days of age. A total of 36,050 records were collected from the movements of 30 young BSs. The analysis of raw data from individual loggers showed that the duration of flight in the immediate vicinity of the nest ranged from 10 to 42 days for individuals, with an average of 21 days. The duration of the flight of young storks was not related to the number of young in the nest (H = 0.68, df = 2, *p* = 0.71). The average daily distance traveled by young birds until they left the nest was 0.44 km (SE = 0.24). The daily sum of flight lengths of individual birds increased with the age of the birds (Figure 2). Once they had acquired the ability to fly, the young birds began to fly longer and longer distances in the days that followed (H = 189.7, df = 4, *p* < 0.001). Individual behaviors were highly variable, with some birds flying much longer distances than most birds. The mean flight length (Spearman’s R_s_ = 0.51, *n* = 1030, *p* < 0.001) and daily sum of flight lengths (Spearman’s R_s_ = 0.53, *n* = 694, *p* < 0.001) increased in proportion to the lengthening of the young storks’ lives (Figure 2 and Figure 3). The relationships found were statistically significant.

In most of the young BSs studied, longer exploratory flights outside the nest stand between 65 and 75 days of age were very weakly marked. Their spatial activity during the first 7–10 days of flight consisted mainly of short flights within 200 m of the nest. Online camera recordings [42] showed that the juveniles immediately returned to the nest at the sight of an adult or the sound of feeding. Only a few days before the start of the migration did some individuals move considerably farther away (the farthest being 8.6 km and 11 km) in search of feeding grounds, but this was incidental, and they eventually returned to or near the nest. This was clearly a response to the abandonment of parental feeding and the need to forage independently. The proportion of time spent in the buffer zone up to 200 m from the nest ranged from 51% to 97% for the 30 BSs studied, with an average of 82% (Table 1). The frequency of use of individual buffers was not related to the number of young in the nest (H = 2.08, df = 2, *p* = 0.35).

The average proportion of time spent by BSs in the periodic protection zones >200–500 m was 2.97%. In contrast, the average proportion of time spent by birds in the buffer zone 500–1000 m from the nest was 1.65% (Table 1). The proportion of time spent further than 1000 m from the nest was 4.41%. The final stage of independence, on the other hand, showed a high (8.02%) proportion of time spent more than 1000 m away (Table 1). However, this was already the case for independent, fully flying individuals and was not significant in terms of the zonal protection of nests. Similarly, the likelihood of BSs being present depended on the distance from the nests (Figure 4). Birds were most often recorded within a 50 m radius of the nest; the farther away from the nest, the lower the probability was of recording the presence of a stork. In a buffer of more than 400 m, the probability of recording a BS did not exceed 0.1. A comparison of the proportion of time spent by birds in each buffer with the proportion of area occupied by the buffer shows that young BSs strongly preferred the area in the immediate vicinity of the nest until independence (χ^2^ = 160.9, df = 4, *p* < 0.001). Despite the fact that it was possible to confirm the statistical significance of the presented relations, it is important to note the high variability in the data collected due to differences in the behaviors of individual birds. This is well illustrated by the large number of outliers in Figure 2 and Figure 3, which show the daily movements of the birds from the time they acquired the ability to fly.

### 3.2. Breeding Phenology

The earliest hatching date for young BSs was the 119th day of the year (29 April), and the latest was the 148th day of the year (28 May). The average hatching date was the 135th day of the year, or 15 May. The hatching date of BS chicks varied according to the geographical location of the nest. In nests further south, the chicks hatched earlier (Spearman’s R_s_ = 0.53, N = 30, *p* = 0.002). In nests further east, the chicks also hatched earlier (Spearman’s R_s_ = −0.46, N = 30, *p* = 0.009, Figure 5).

The first storks left the nest on day 212 of the year (31 July). The latest departure was recorded on day 245 of the year (2 September). The average date of departure of a young BS in Poland was day 222 of the year, i.e., 10 August. Nests located further south were left earlier by young birds (Spearman’s R_s_ = 0.49, N = 30, *p* = 0.006). Nests located further west were left later (Spearman’s R_s_ = −0.46, N = 30, *p* = 0.009, Figure 5).

Birds left the nest between 73 and 108 days of age, with an average of 87 days of age. Values below 75 days of age were recorded for frightened birds. The age at which birds left the nest was not correlated with the geographical location of the nest, either for longitude (Spearman’s R_s_ = 0.185, N = 30, *p* = 0.324) or latitude (Spearman’s R_s_ = −0.186, N = 30, *p* = 0.328). The number of young in the brood was not significant for the time of nest departure (H = 0.74, df = 2, *p* = 0.69).

After the 70th day of life, young storks usually spent the night outside the nest in the stand at a distance of 200–250 m from the nest. An attempt was made to estimate the age at which young birds should naturally leave the nest. Data were compared for consecutive days in the life of the young, whether they stayed in or left the nest. The probabilistic logistic regression model used indicated the 85th day of life as the cut-off point for the probability of more than 50% of young BSs leaving the nest (Figure 6, Table 2).

### 3.3. Resting Places Near Nests

From the data collected by the loggers, it was possible to identify the most frequently visited sites near the nest by the young birds, i.e., nodal points. The vast majority of these points were in the forest surrounding the nest (81%), with the remainder in non-forest areas (19%). A total of 90% of the birds had one to three nodal points around the nest, and only 9% of the birds had four points. One young stork regularly used five points in the surrounding revetment. Most of the points were in pine stands (80%), and only 20% were in deciduous stands. Forest age ranged from 5 to 124 years, with an average of 75 years. Nodal points were located between 224 and 936 m from the nest, with an average of 450 m. Moreover, 73% of the nodal points were within 500 m of the nest, with only one in four resulting from flights more than 500 m from the nest.

## 4. Discussion

The identification of bird nesting habitats is a prerequisite for their effective protection [2,4,43]. Verifying conservation methods for rare species using telemetry data can serve to increase the effectiveness of conservation efforts [34,44]. The use of nest protection buffer zones seems to be an appropriate strategy for the conservation of large endangered bird species. The definition of such zones is often based on expert knowledge [45]. For obvious reasons, it is difficult to obtain scientific data for zone delimitation based on the experimental measurement of flushing distance and its effect on breeding success (except Margalida et al. [46]). This paper proposes a different approach by defining a radius from the nest where young BSs prepare to migrate. Our research is the first attempt to use telemetry surveys to study the behavior of BSs in the early post-breeding season. We chose typical nests and territories for the Polish BS population, where the nests on oaks are preferable [47]. The results show that until the nest was completely abandoned, the zone within 200 m of the nest was the most heavily used. Birds spent more than 80% of their time in this area, and the probability of finding them here was in the range of 0.2–0.5. Birds spent an average of just over 6% of their time in the zone above 500 m. Seventy-three percent of the nodal points were within a radius of less than 500 m from the nest. These were most likely resting areas where birds rested while learning to fly and explored the nest area as they became independent. The area provided a space for young birds to find their way back to the nest and for adults to continue feeding. Young storks are very attached to the immediate vicinity of the nest. During the first period of independence, BSs are largely honing their flying skills and using the immediate nest area as their main area of activity. The success of the brood depends largely on the peace and quiet around the nest [48]. In light of our results, while the 500 m protection buffer zones around nests used in Poland seem large, they provide a fully peaceful place for the undisturbed growth and development of young storks. In addition, they protect the nesting habitat from the negative impacts of forest management [49].

In the present study, young storks left the nest for the first time between 70 and 108 days of age, with a mean age of 87 days, whereas Cramp and Simmons [15] found that nest departure occurs between 63 and 71 days of age. In central Poland, however, storks were recorded in the nest at 77 and 87 days of age [50], which is closer to the results obtained in the present study using telemetry. Observational data on BSs from Belarus indicate nest abandonment at 72–75 days of age [51]. Larue et al. [36] showed that after the first flight, young BSs are connected to the nest site for another 30 days, which is very similar to our findings. Depending on the methodology used, data on earlier nest abandonment may refer to first flights and observations of empty nests rather than nest abandonment for migration. We confirmed that the exploratory range of young birds correlated with their age. The older the bird, the longer the distance traveled and the more often it attempted to fly. There was considerable variability in this trend, with individuals as young as 75 days flying longer distances up to two weeks earlier than average. The longest-staying birds did not take longer flights around the nest until they were over 100 days old. The average time between the first flight from the nest and the start of migration was 21 days. For long-distance migrants, such as BSs, the period before the start of migration has a crucial influence on the condition of the bird and may affect the survival rate of migrating birds [33,37]. The foraging and residence of juvenile BSs have not been observed together, as shown by a telemetry study in France [36]. Temporary protection of breeding sites should last from the arrival of the adult birds until the young have left the nest. In the case of the black stork, the egg-laying and incubation time is about 40 days. Before that, the birds spend some time after arrival choosing a breeding site and preparing their nest. Our data show that the young are in the nest and the immediate area for an average of 87 days and a maximum of 108 days. The Polish system, with 168 days of extended protection in a wider 500 m buffer, could be improved by adding time provisions based on the arrival date and beginning of breeding of the BS pairs.

The dates of hatching and nest abandonment appeared to be related to the geographical location of the nest. Nests located further south, closer to the wintering grounds, were characterized by significantly earlier hatching and nest-abandonment dates. It was also found that birds in the east of the country hatched and left the nest earlier than those in the west. This may indicate that the adult birds of the populations studied move mainly along the eastern migration route, which, after bypassing the Carpathian Arc, allows earlier occupation of breeding territories in the southeastern part of Poland. Similarly, BSs fly to other countries in the region [52]. The data may be somewhat biased by the fact that nests from the western part of Poland came mainly from the north of the country. Importantly, the age of chicks leaving the nest was not correlated with latitude or longitude. This suggests that young birds require a similar amount of time to develop and prepare for migration. Young birds migrate independently of their parents, so there is no mechanism for adults to force young birds to leave the nest [36]. In addition, the results suggest that birds from the north of the country simply have less time to successfully breed. The differences in the timing of hatching of juvenile BSs in different parts of Poland that we have shown confirm the need to review the date of commencement of periodic protection so that it corresponds to the biology of the species. Data from Poland at the beginning of the century indicated a 30 March arrival date for BS to the west of the country and of April 6 to central Poland [53,54]. Current data from the last decade [55] collected by birders across the country indicate an acceleration of arrival dates. Between 2015 and 2024, the first BSs in Poland were observed between 14 and 26 March; on average, it was 19 March in the central part of the country. This result suggests that the period of nest protection in a wider radius should be extended and take effect as early as 1 March to ensure that the birds are calm around their nests when they return from their wintering grounds. Such an arrangement would provide sufficient protection, taking into account the differences in bird arrivals in different parts of the country.

These telemetry results suggest the need for the protection of BS nesting sites from the start of the breeding season in March until the end of August to ensure that these rare birds remain undisturbed and to guarantee the absence of dangerous interactions with humans, which can greatly reduce breeding success and, in the long term, harm the entire population. The effectiveness of protective buffers zone for preserving nesting trees has already been proven [22]. The high value of BS protection zones for protecting forest biodiversity has also been proven [56]. The current work demonstrates the great importance of the buffer zone method for ensuring that young birds have the right conditions to grow and learn to fly. Reducing anthropogenic pressure during black stork nesting, especially during the phase of young BSs learning flight, is crucial for the successful completion of the nesting and the departure of the birds.

## 5. Conclusions

The use of telemetry to study BSs in the post-breeding period provided new information on the pattern of young birds’ independence. It was possible to determine the basic parameters of breeding phenology, such as the age of making the first flight attempts at an average of 65 days and the final departure from the nest at an average of 87 days. The use of space during this period was analyzed. The birds make the greatest use of a 500 m radius around the nest. In this zone, the birds refine their flying skills, explore the surroundings, and gradually become independent. The immediate vicinity of the nest and selected roosts are crucial. The data collected suggest that there are differences in the timing of hatching depending on the geographical location of the nest. This may be related to the migration route of adult birds from wintering grounds and the timing of their arrival at nesting sites. The results confirm the validity of the establishment of protection zones of 500 m around nests and the need to maintain them from the start of the breeding season in March until the end of August.

## Figures and Tables

**Figure 1 animals-14-01558-f001:**
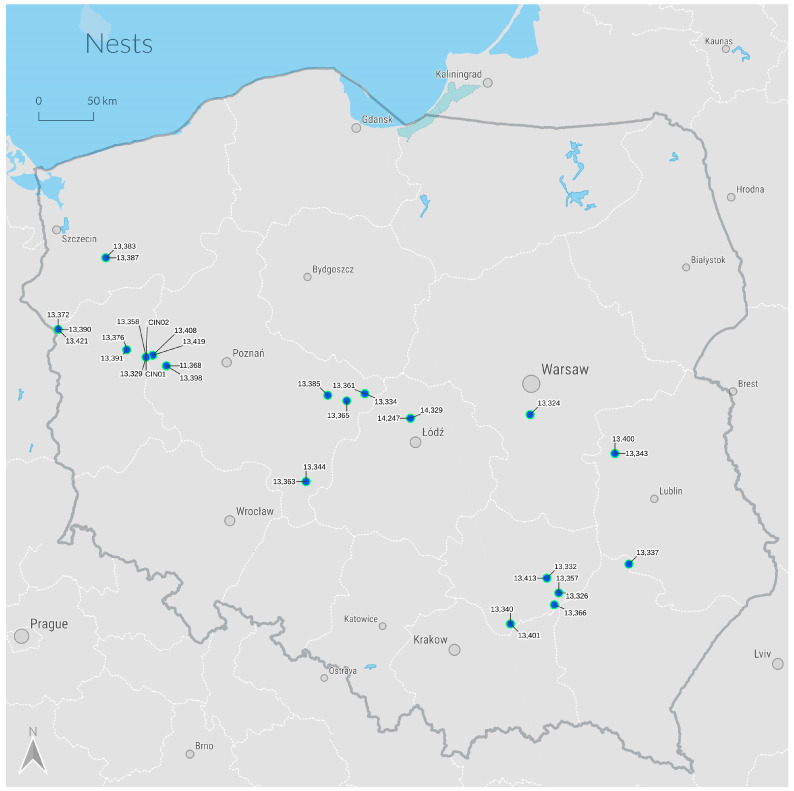
Map of nests selected for the installation of bird transmitters with the transmitters’ identification numbers.

**Figure 2 animals-14-01558-f002:**
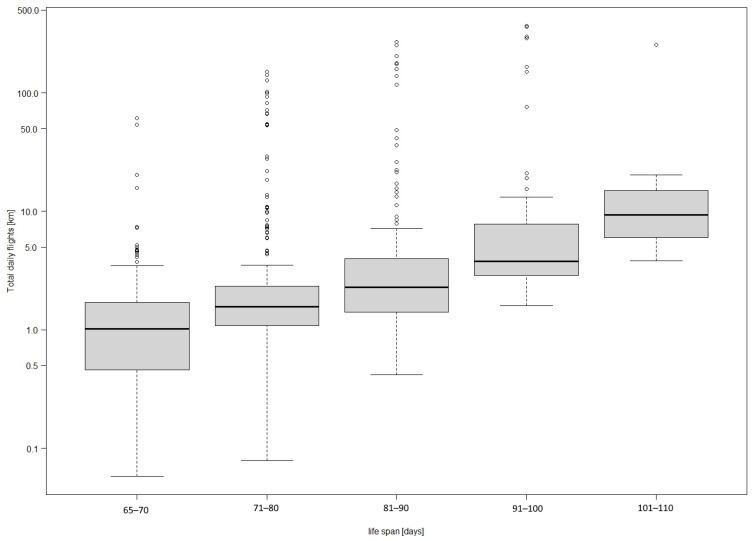
Changes in total flight length over the stork’s life span. Boxes with different letters are significantly different at the *p* < 0.01 level. Distance of daily flights on a logarithmic scale.

**Figure 3 animals-14-01558-f003:**
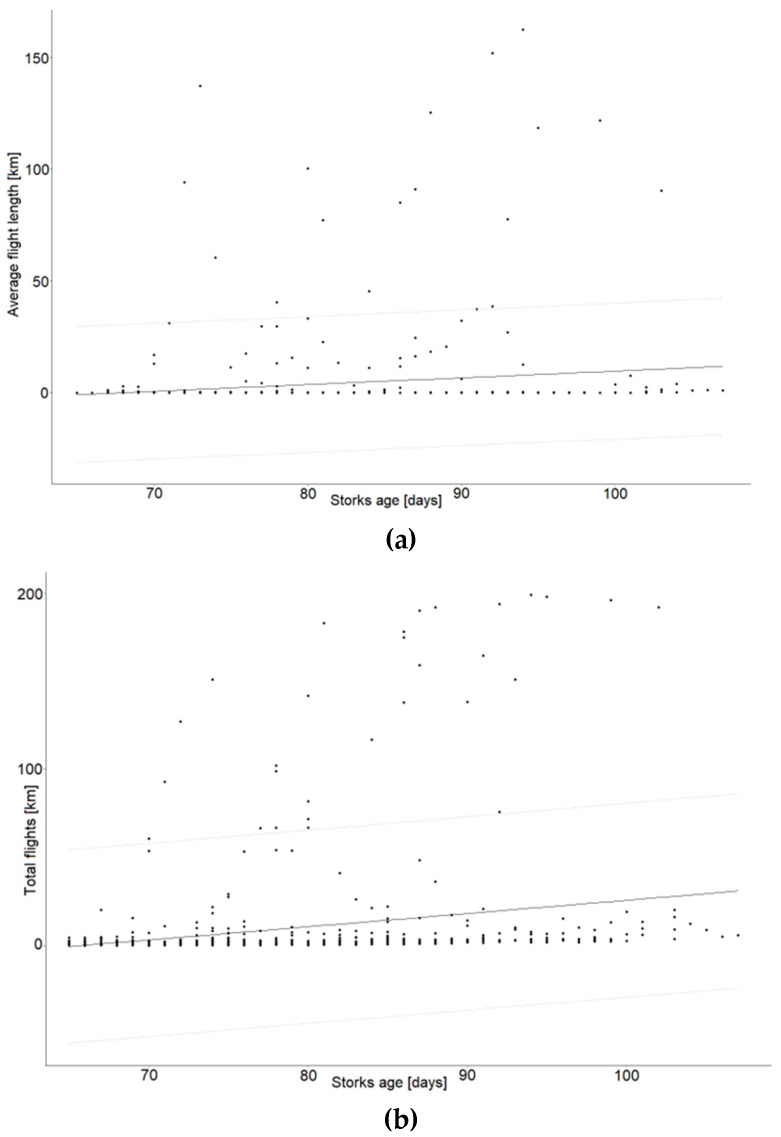
Correlation of average daily flight length (**a**) and daily total flights (**b**) with age of young storks, with 95% prediction intervals. Distances above 200 km have been omitted for better readability of the graph.

**Figure 4 animals-14-01558-f004:**
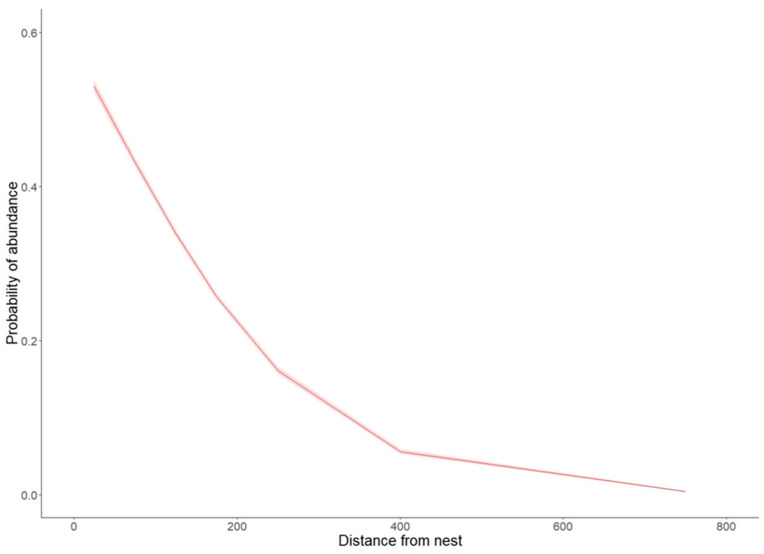
Probability of stork presence depending on distance from the nest according to logistic regression model, with 95% confidence intervals (intercept = 0.73, SE = 0.03, z = 29.06, *p* < 0.0001; distance from the nest = −0.007, SE = 0.0001, z = −63.23, *p* < 0.0001).

**Figure 5 animals-14-01558-f005:**
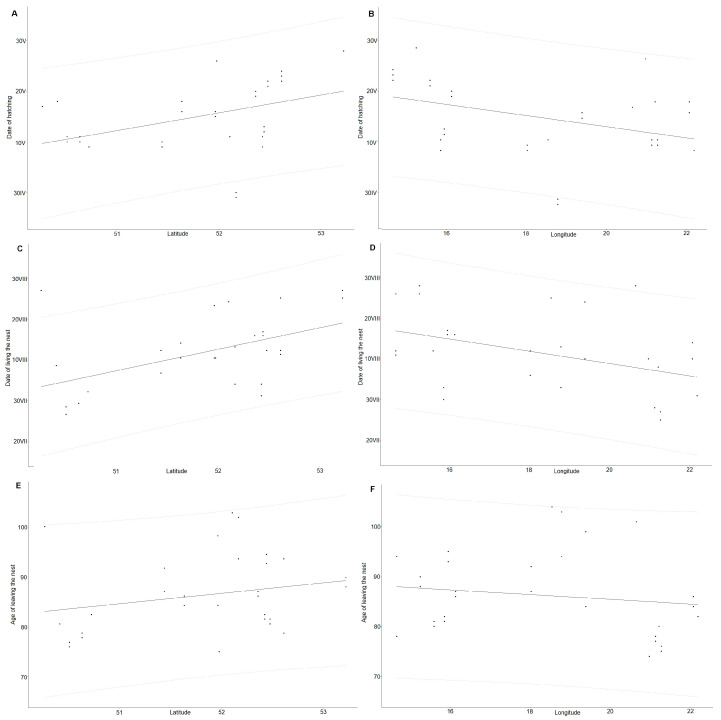
Correlation of young black storks’ hatching day (**A**,**B**), young black storks’ nest-leaving day (**C**,**D**), and black storks’ nest-leaving age (**E**,**F**) with longitude and latitude, with 95% prediction intervals.

**Figure 6 animals-14-01558-f006:**
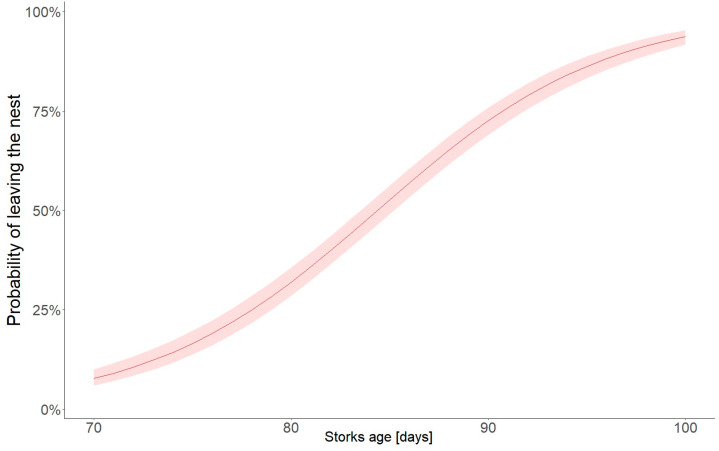
Probability of young storks leaving the nest depending on age, with 95% confidence intervals.

**Table 1 animals-14-01558-t001:** Time spent by surveyed individuals of black storks at distance intervals.

	<200 m		200–500 m		500–1000 m		>1000 m		Rejected Data	
Individual	Time	%	Time	%	Time	%	Time	%	Time	%
13385	742:09:56	96.89	00:19:57	0.04	00:49:58	0.11	01:10:05	0.2	21:30:12	2.81
13408_1	529:06:08	69.07	71:19:59	9.31	32:41:02	4.27	16:25:46	2.1	116:27:09	15.20
13419_1	533:53:41	69.98	67:59:23	8.91	23:59:31	3.14	11:59:49	1.6	125:06:18	16.40
11368_2	713:14:37	93.11	08:13:42	1.07	01:10:47	0.15	13:42:32	1.8	29:38:20	3.87
13398_2	716:00:40	93.47	04:00:02	0.52	00:00:00	0.00	14:00:14	1.8	31:59:16	4.18
13376_3	654:58:28	85.51	23:00:10	3.00	00:00:00	0.00	61:00:02	8.0	27:00:57	3.53
13391_3	603:50:04	79.81	55:25:34	7.33	02:50:48	0.38	46:23:49	6.1	48:04:12	6.35
13372_4	715:01:51	93.35	00:00:00	0.00	06:00:32	0.78	11:00:00	1.4	33:57:26	4.43
13390_4	391:52:04	51.10	74:56:10	9.77	06:06:02	0.80	204:47:09	26.7	89:08:32	11.62
13421_4	728:59:30	95.07	04:00:25	0.52	00:00:00	0.00	14:50:02	1.9	18:59:23	2.48
13383_5	705:57:09	92.16	00:01:26	0.00	02:59:32	0.39	14:00:06	1.8	43:01:07	5.62
13387_5	715:00:17	93.24	00:00:00	0.00	00:00:00	0.00	11:50:24	1.5	39:59:40	5.22
13329_1	59:00:16	42.15	13:00:01	9.29	24:59:59	17.86	26:00:39	18.58	16:58:59	12.13
13358_1	257:00:54	64.09	65:00:24	16.21	02:00:07	0.50	01:00:01	0.25	75:58:34	18.95
CIN01_1	325:35:00	82.22	09:18:45	2.35	00:17:44	0.07	44:05:32	11.13	16:43:05	4.22
CIN02_1	56:02:13	29.34	00:00:00	0.00	00:00:00	0.00	132:25:08	69.33	02:31:54	1.33
13324_2	219:59:09	92.43	00:00:00	0.00	00:00:00	0.00	15:00:03	6.30	03:00:22	1.26
14274_3	439:01:03	92.23	05:00:08	1.05	00:00:00	0.00	12:59:49	2.73	18:58:31	3.99
14329_3	688:32:55	82.56	21:29:30	2.58	65:59:15	7.91	13:00:00	1.56	44:58:09	5.39
13337_4	339:59:04	79.25	05:00:59	1.17	11:00:04	2.56	38:00:12	8.86	34:58:53	8.15
13334_5	574:58:45	80.30	18:59:52	2.65	35:00:21	4.89	07:59:39	1.12	79:01:13	11.04
13361_5	708:28:39	75.89	12:01:43	1.29	74:27:50	7.98	14:59:19	1.61	123:32:35	13.23
13340_6	743:25:09	83.96	25:58:55	2.93	02:00:54	0.23	35:00:04	3.95	79:04:02	8.93
13401_6	818:57:04	79.55	22:28:36	2.18	15:00:14	1.46	64:31:09	6.27	108:33:11	10.54
13344_7	443:59:27	81.02	12:00:00	2.19	00:00:00	0.00	47:59:40	8.76	44:00:26	8.03
13363_7	587:58:18	87.89	11:59:52	1.79	00:00:00	0.00	13:08:23	1.96	55:52:42	8.35
13343_8	422:58:56	80.57	03:59:57	0.76	40:30:46	7.72	17:00:02	3.24	40:29:59	7.71
13400_8	431:54:54	90.75	01:00:00	0.21	00:00:00	0.00	30:00:04	6.30	13:00:10	2.73
13366_9	325:00:38	85.53	12:00:32	3.16	00:59:44	0.26	12:00:41	3.16	29:59:01	7.89
13332_10	229:58:55	69.91	02:00:02	0.61	03:00:26	0.91	10:59:55	3.34	83:00:06	25.23
13413_10	230:00:10	75.41	07:58:53	2.62	13:00:46	4.27	11:00:15	3.61	42:59:49	14.10
13326_11	67:59:58	24.29	120:00:20	42.86	04:00:24	1.43	20:00:14	7.14	67:58:55	24.28
13357_11	111:58:58	42.91	75:59:18	29.12	10:00:34	3.84	11:00:19	4.22	51:59:49	19.92
13365_12	808:01:31	84.74	49:00:11	5.14	07:29:19	0.79	30:30:41	3.20	58:28:38	6.13

**Table 2 animals-14-01558-t002:** Logistic model results for probability of young black storks leaving the nest (binomial error distributed with logit link).

Fixed Effect Parameter	Estimate	Std. Error	*z* Value	*p*
Intercept	−14.574	0.74	−19.75	<0.0001
Day of life	0.173	0.009	19.86	<0.0001
AIC	1165.3			

## Data Availability

Data are contained within the article.

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
