# Peer review of "Are Current Protection Methods Ensuring the Safe Emancipation of Young Black Storks? Telemetry Study of Space Use by Black Storks (Ciconia nigra) in the Early Post-Breeding Period"

_animals, 2024, doi:10.3390/ani14111558_

Round 1

Reviewer 1 Report

Comments and Suggestions for Authors

General comments

This article studies the space use of 34 fledgling black storks by GPS/GPRS transmitters using LTE/GSM technology and photo traps cameras in western and central-eastern of Poland. “The objective of the study was to investigate the spatial behaviour of young black storks in the stands surrounding their nests during the post-nesting period, from their first flight until they leave the nest.  The study hypothesized that young black storks, before they definitively leave the nest and undertake migration, make practice flights within a radius of 500 meters from the nest and rarely move further away. As the birds get older, the radius of the area used will increase. In addition, the study was designed to verify the validity of designating a 200-meter year-round protection zone and a 500-meter or periodic zone”.

In general, the topic of the article is really interesting. There are only few studies of post-breeding movements of fledgling black storks as the article points out, and this topic implies some conservation issues for the species and its habitat management. So, the topic would provide a new and novel knowledge of this species from my point of view. However, the hypothesis, some methods, and data analysis are not clear to me, and I am not really convincing on the approach of this research.

First of all, from my personal opinion, the hypothesis is weak as it is an obviousness. The authors hypothesis is: “young black storks, before they definitively leave the nest and undertake migration, make practice flights within a radius of 500 meters from the nest and rarely move further away. As the birds get older, the radius of the area used will increase. In addition, the study was designed to verify the validity of designating a 200-meter year-round protection zone and a 500-meter or periodic zone”. Obviously, all young black storks will performance flies within a short radius from the nest in their first flights, actually, the study finds distances below 200 m during the first 7-10 days after the first flight of the juveniles, and obviously the distance will increase while the bird is older. It seems the study’s aim is focused on validate what it is established as protection zones (200 m around the nest during the breeding season and 500 m across the year). Moreover, the study tries to analyse other issues such as the influence of the brood sizes the period of the independency of the juveniles from the nesting areas, the use of tree species after the first flight, for example, but these issues are not included in the introduction/hypothesis or the aims that the study expects to explore. I miss a clear stand of the aims or strong hypotheses of the study.

Secondly, I am not convincing on the methods and data analysis used for the study:

1)     The authors write “The data provided by the photo loggers enabled us to accurately determine the date of the fledglings' first flights, record the moment of departure from the nest, and evaluate whether it was due to predation or human activity”. If the juveniles are tagged by GPS/GPRS transmitters, there is no need of using photo cameras to determine the data of “the fledglings' first flights, record the moment of departure from the nest”, just the authors could check the data provided by the transmitters. Also, the first flight from the nest of fledgling storks luckily is near to a branch, even in the same tree. In this case, do the authors count this as the “first flight” or not? If the first flight considered is the one recorded by the camera, maybe you consider this first movement, even if this movement is out of the camara’s field of view.  Maybe, I am wrong and I am missing something, I do not feel confident with this approach

2)     The authors write “The data covers the life span of young storks from the 65th day of the chicks' life, which is when they take their first flight outside the nest, to the first day of migration. The age of the chicks was determined based on the development of their plumage during ringing and transmitter placement. The hatching date of young black storks was reconstructed with an accuracy of 2 days based on the age of the chicks at the time of ringing. These data were then used to calculate the age of the birds on the day they left the nest, and were subsequently used in statistical analyses”. As far as it is written, I understand the analysis started form the 65th day of the chick life for all tagged fledglings, like all fledgling black storks start the first flight out of the nest at the age of 65 days (however, the authors wrote that the “birds left the nest between 67 and 108 days of age”, line 252). This is not real and right, there is a great variation among fledglings for this (as they found in the results). On the other hand, they estimated the age based on the plumage, that vary more than the length of the body. There are formulas to better estimation of the chick based on the body:

Age estimation

KAMINSKI, MACIEJ; JANIC, BARTOSZ; MARSZAL, LIDIA; BANBURA, JERZY; and ZIELINSKI, PIOTR (2018) "Age estimation of black stork (Ciconia nigra) nestlings from wing, bill, head, and tarsus lengths at the time of ringing," Turkish Journal of Zoology: Vol. 42: No. 1, Article 17. https://doi.org/10.3906/zoo-1702-42

Available at: https://journals.tubitak.gov.tr/zoology/vol42/iss1/17

3)     The authors write regarding the departing day: “The day on which the birds flew more than 10km from the nest and did not return for the night was considered as the date of departure from the nest”. The authors do not have any reason why the decided a distance more than 10 km, is this based on any biological evidence on the species or any previous study on birds? The authors should explain why they decide such distance and based on.

4)     The authors point out that there are no many studies on movement black storks during the breeding season (21), but there are for raptors (1,5,22). This is right. Most of the studies on movement use kernel and minimum polygon convex methods to study the use of space on the tagged birds, I am missing why the authors do not explain why they do not use this approach and chose concentric circles with different radius, what is the advantage of this method respect to the others. Studies based on kernels and MPC give more precise distances and use of space based on the data than a pre-fixed concentric circles.

5)     Because I am doubting on the used methods, and I am not clear, I do not mention the statistics used for the analysis.

Thirdly, I think the introduction (see concrete comments) and the discussion are not clear, precise and lack of a strong framework. For example, the paragraph of the discussion, lines 321-340. The authors are speculating too much regarding the migration flyway of the breeders. There are not so many records of Polish black storks crossing the strait of Gibraltar (there are few records, but we cannot ensure the western Polish population migrate through the western flyway and this is why they have a late phenology without tagging the breeders).

 Fourthly, the authors are missing relevant references on the topic and the species. I think the authors should check more bibliography regarding the post-breeding period (as the title indicates) to be taken into account for this study. There are a lot of literature for other groups of birds (you can search in different sources), even a study on black stork: Luis Santiago Cano, Cláudia Franco, Guillermo Doval, Alejandro Torés, Isidoro Carbonell, José Luis Tellería "Post-Breeding Movements of Iberian Black Storks Ciconia nigra as Revealed by Satellite Tracking ," Ardeola, 60(1), 133-142, (1 June 2013). Moreover, there are a couple of new papers they may consider for this study: Fisel, F., Heine, G., Rohde, C. et al. Influence of age on spatial and temporal migratory patterns of Black Storks from Germany. J Ornithol (2024). https://doi.org/10.1007/s10336-024-02170-3 and Väli Ü, Strazds M, Kaldma K, Treinys R. Low juvenile survival threatens the Black Stork Ciconia nigra in northern Europe. Bird Conservation International. 2024;34:e10. doi:10.1017/S0959270924000042

Fifthly, reading the paper, I feel that the authors inserted phrases in sections that are not the most appropriate place (please, see concrete comments).

Finally, I am not native English speaker, but the English writing of the article has been really difficult to understand to me.

Concrete comments:

Line 1-2:

Title: I would review the title, it is not a “telemetry studies”, in fact it is only a single “study”, and the use of space is focussed on the emancipation period of the fledglings’ black storks and not for all post-breeding period (mean, until they start the migration). Moreover, the study includes some aspects of the phenology of the species and the effect of the latitude and longitude.

Line 15-16:

The research presented here is an attempt to use GPS loggers…”.

I do not understand the meaning of “attempt” in the sentence. I would remove “is an attempt” and directly say: “The research uses GPS…..

Line 29-30:

A shorter migration route from the wintering grounds allowed the young to reach the nest earlier and to begin to fledge…”.

This sentence is not clear to me. Shorter migration from the wintering grounds. I do not understand the meaning as the breeders are the ones that reach the nest from the wintering grounds.

Line 36-37

The rapid development of telemetry in recent years has allowed its wider use in the study of the biology of mainly….

The telemetry is used since the 60´s last century, and the PTT transmitters since the 90´s in black storks, I think is not “in recent years anymore.

Line 46-47

The range of the BS extends to the temperate regions of Europe and the western part of 46 Asia [13]

The range of the black stork is almost the whole Palearctic (Iberian Peninsula up to Mongolia and Korean peninsula) and south Africa as breeding range, but include all sub-Saharan region and Indian sub-continent.  

Line 53-56

“…. make it necessary to better understand the detailed aspects of breeding biology that influence the health of black stork populations. Loggers and photo-traps are good tools for such studies…”.

I do not understand the deep meaning of this sentence, the relationship of the use of photo-traps and loggers, the breeding biology and the health of black populations. Maybe the authors could re-phrase the sentence for a better and clear understanding for the readers.

Line 62-64:

Chevallier et al. [24] found it unlikely that storks foraged near the nest just before migratory flights. However, our initial field observations suggest that young birds stay in stands close to the nest for some time”. 

2 issues: 1) I do not find any reference on this in Chevallier et al. [24], there is no any “nest” word in this study or reference on this topic when Chevallier et al. commented about the breeding areas; 2) is this sentence appropriate as it is written for the introduction? From my opinion, this sentence could be more appropriate in the discussion, when the authors have disclosed the results.

Line 85-86:

No negative effects on adult storks were observed”.

I think this sentence is more appropriate to be included in the results rather than methods.

Line 94-95:

A total of 36,050 records were collected from the movements of these individuals”.

I think this sentence is more appropriate to be included in the results rather than methods.

In the results, the authors provide a number of average, but they do not give any standard deviation or range of the average

Comments on the Quality of English Language

I am not native English speaker, so I do not feel good enough to comment on this regard, but there are parts of the article that it is difficult to understand to me because of the English writing of this article. 

Author Response

This article studies the space use of 34 fledgling black storks by GPS/GPRS transmitters using LTE/GSM technology and photo traps cameras in western and central-eastern of Poland. “The objective of the study was to investigate the spatial behaviour of young black storks in the stands surrounding their nests during the post-nesting period, from their first flight until they leave the nest.  The study hypothesized that young black storks, before they definitively leave the nest and undertake migration, make practice flights within a radius of 500 meters from the nest and rarely move further away. As the birds get older, the radius of the area used will increase. In addition, the study was designed to verify the validity of designating a 200-meter year-round protection zone and a 500-meter or periodic zone”.

In general, the topic of the article is really interesting. There are only few studies of post-breeding movements of fledgling black storks as the article points out, and this topic implies some conservation issues for the species and its habitat management. So, the topic would provide a new and novel knowledge of this species from my point of view. However, the hypothesis, some methods, and data analysis are not clear to me, and I am not really convincing on the approach of this research.

Answer: Our goals have been reformulated to be more clear (Line 101-111).

First of all, from my personal opinion, the hypothesis is weak as it is an obviousness. The authors hypothesis is: “young black storks, before they definitively leave the nest and undertake migration, make practice flights within a radius of 500 meters from the nest and rarely move further away. As the birds get older, the radius of the area used will increase. In addition, the study was designed to verify the validity of designating a 200-meter year-round protection zone and a 500-meter or periodic zone”. Obviously, all young black storks will performance flies within a short radius from the nest in their first flights, actually, the study finds distances below 200 m during the first 7-10 days after the first flight of the juveniles, and obviously the distance will increase while the bird is older. It seems the study’s aim is focused on validate what it is established as protection zones (200 m around the nest during the breeding season and 500 m across the year). Moreover, the study tries to analyse other issues such as the influence of the brood sizes the period of the independency of the juveniles from the nesting areas, the use of tree species after the first flight, for example, but these issues are not included in the introduction/hypothesis or the aims that the study expects to explore. I miss a clear stand of the aims or strong hypotheses of the study.

Answer: We changed the introduction in the manuscript. We added more information about the status of the black stork, the decline in its numbers (line 50-55), and conservation efforts (line 55-67). We highlighted the need to evaluate the effectiveness of the conservation methods that are in place (line 72-77). We added clear aims of the work (line 101-111).

Secondly, I am not convincing on the methods and data analysis used for the study:

  • The authors write “The data provided by the photo loggers enabled us to accurately determine the date of the fledglings' first flights, record the moment of departure from the nest, and evaluate whether it was due to predation or human activity”. If the juveniles are tagged by GPS/GPRS transmitters, there is no need of using photo cameras to determine the data of “the fledglings' first flights, record the moment of departure from the nest”, just the authors could check the data provided by the transmitters. Also, the first flight from the nest of fledgling storks luckily is near to a branch, even in the same tree. In this case, do the authors count this as the “first flight” or not? If the first flight considered is the one recorded by the camera, maybe you consider this first movement, even if this movement is out of the camara’s field of view. Maybe, I am wrong and I am missing something, I do not feel confident with this approach

Answer: We explained the use the camera traps (line 113-128). We clarified the day of nest departure – as the day when young bird stop returning to the nest (line 158-160). We defined first flight as movement minimum 50 meters from the nest (line 151-152).

2)     The authors write “The data covers the life span of young storks from the 65th day of the chicks' life, which is when they take their first flight outside the nest, to the first day of migration. The age of the chicks was determined based on the development of their plumage during ringing and transmitter placement. The hatching date of young black storks was reconstructed with an accuracy of 2 days based on the age of the chicks at the time of ringing. These data were then used to calculate the age of the birds on the day they left the nest, and were subsequently used in statistical analyses”. As far as it is written, I understand the analysis started form the 65th day of the chick life for all tagged fledglings, like all fledgling black storks start the first flight out of the nest at the age of 65 days (however, the authors wrote that the “birds left the nest between 67 and 108 days of age”, line 252). This is not real and right, there is a great variation among fledglings for this (as they found in the results). On the other hand, they estimated the age based on the plumage, that vary more than the length of the body. There are formulas to better estimation of the chick based on the body:

Age estimation

KAMINSKI, MACIEJ; JANIC, BARTOSZ; MARSZAL, LIDIA; BANBURA, JERZY; and ZIELINSKI, PIOTR (2018) "Age estimation of black stork (Ciconia nigra) nestlings from wing, bill, head, and tarsus lengths at the time of ringing," Turkish Journal of Zoology: Vol. 42: No. 1, Article 17. https://doi.org/10.3906/zoo-1702-42

Available at: https://journals.tubitak.gov.tr/zoology/vol42/iss1/17

Answer: We corrected the sentence (line 148-149). 65th day was chosen, as the result of birds behaviour. Any of the birds had not fly before 65th day of life. We added more precise description of assessing the age of birds, in according to the paper of Kamiński et al. (Line 152-157)

  • The authors write regarding the departing day: “The day on which the birds flew more than 10km from the nest and did not return for the night was considered as the date of departure from the nest”. The authors do not have any reason why the decided a distance more than 10 km, is this based on any biological evidence on the species or any previous study on birds? The authors should explain why they decide such distance and based on.

Answer: We clarified that the moment of leaving the nest was the day of departure followed by no return to the nest, and it was always more than 10 km, but following the reviewer's suggestion we exclude distance from the definition (line 158-160).

  • The authors point out that there are no many studies on movement black storks during the breeding season (21), but there are for raptors (1,5,22). This is right. Most of the studies on movement use kernel and minimum polygon convex methods to study the use of space on the tagged birds, I am missing why the authors do not explain why they do not use this approach and chose concentric circles with different radius, what is the advantage of this method respect to the others. Studies based on kernels and MPC give more precise distances and use of space based on the data than a pre-fixed concentric circles.

Answer: As the aim of the study was to assess the effectiveness of the protective buffer zone (determined as a radius from the nest), it was more convenient to take an approach that assessed the presence of birds within buffer distances from the nest rather than based on estimates of the polygon used by the young birds. Wi provide explanation in lines 173-176.

  • Because I am doubting on the used methods, and I am not clear, I do not mention the statistics used for the analysis.

Answer: We very much regret that we did not get support for this.

Thirdly, I think the introduction (see concrete comments) and the discussion are not clear, precise and lack of a strong framework. For example, the paragraph of the discussion, lines 321-340. The authors are speculating too much regarding the migration flyway of the breeders. There are not so many records of Polish black storks crossing the strait of Gibraltar (there are few records, but we cannot ensure the western Polish population migrate through the western flyway and this is why they have a late phenology without tagging the breeders).

Answer: Following the reviewer's suggestion, we have restructured the introduction and discussion. In particular, we have reworked the disputed passage to make it less speculative (line 376-384). 

 Fourthly, the authors are missing relevant references on the topic and the species. I think the authors should check more bibliography regarding the post-breeding period (as the title indicates) to be taken into account for this study. There are a lot of literature for other groups of birds (you can search in different sources), even a study on black stork: Luis Santiago Cano, Cláudia Franco, Guillermo Doval, Alejandro Torés, Isidoro Carbonell, José Luis Tellería "Post-Breeding Movements of Iberian Black Storks Ciconia nigra as Revealed by Satellite Tracking ," Ardeola, 60(1), 133-142, (1 June 2013). Moreover, there are a couple of new papers they may consider for this study: Fisel, F., Heine, G., Rohde, C. et al. Influence of age on spatial and temporal migratory patterns of Black Storks from Germany. J Ornithol (2024). https://doi.org/10.1007/s10336-024-02170-3 and Väli Ü, Strazds M, Kaldma K, Treinys R. Low juvenile survival threatens the Black Stork Ciconia nigra in northern Europe. Bird Conservation International. 2024;34:e10. doi:10.1017/S0959270924000042

Answer: These papers coincided in time with the submission of this manuscript. We have now added them to the literature review. Thank you for suggesting them.

Fifthly, reading the paper, I feel that the authors inserted phrases in sections that are not the most appropriate place (please, see concrete comments).

Answer: We corrected some mistakes.

Finally, I am not native English speaker, but the English writing of the article has been really difficult to understand to me.

Answer: The manuscript has undergone English language editing by MDPI. The text has been checked for correct use of grammar and common technical terms, and edited to a level suitable for reporting research in a scholarly journal.

Concrete comments:

Line 1-2:

Title: I would review the title, it is not a “telemetry studies”, in fact it is only a single “study”, and the use of space is focussed on the emancipation period of the fledglings’ black storks and not for all post-breeding period (mean, until they start the migration). Moreover, the study includes some aspects of the phenology of the species and the effect of the latitude and longitude.

Answer: We have rephrased the title of the manuscript to better match the content of the work (Line 1-4)

Line 15-16:

The research presented here is an attempt to use GPS loggers…”.

I do not understand the meaning of “attempt” in the sentence. I would remove “is an attempt” and directly say: “The research uses GPS…..

Answer: Corrected. Line 18-19

Line 29-30:

A shorter migration route from the wintering grounds allowed the young to reach the nest earlier and to begin to fledge…”.

This sentence is not clear to me. Shorter migration from the wintering grounds. I do not understand the meaning as the breeders are the ones that reach the nest from the wintering grounds.

Answer: We have rephrased the sentence to make it clearer and to avoid a mental shortcut. (line 32-33)

Line 36-37

The rapid development of telemetry in recent years has allowed its wider use in the study of the biology of mainly….

The telemetry is used since the 60´s last century, and the PTT transmitters since the 90´s in black storks, I think is not “in recent years anymore.

Answer: We have rephrased the sentence to make it clearer  (Line 39-41)

Line 46-47

The range of the BS extends to the temperate regions of Europe and the western part of 46 Asia [13]

The range of the black stork is almost the whole Palearctic (Iberian Peninsula up to Mongolia and Korean peninsula) and south Africa as breeding range, but include all sub-Saharan region and Indian sub-continent. 

Answer: Corrected (Line  48-51)

Line 53-56

“…. make it necessary to better understand the detailed aspects of breeding biology that influence the health of black stork populations. Loggers and photo-traps are good tools for such studies…”.

I do not understand the deep meaning of this sentence, the relationship of the use of photo-traps and loggers, the breeding biology and the health of black populations. Maybe the authors could re-phrase the sentence for a better and clear understanding for the readers.

Answer: We have rephrased the sentence to make it clearer  (Line 81-85)

Line 62-64:

Chevallier et al. [24] found it unlikely that storks foraged near the nest just before migratory flights. However, our initial field observations suggest that young birds stay in stands close to the nest for some time”. 

2 issues: 1) I do not find any reference on this in Chevallier et al. [24], there is no any “nest” word in this study or reference on this topic when Chevallier et al. commented about the breeding areas; 2) is this sentence appropriate as it is written for the introduction? From my opinion, this sentence could be more appropriate in the discussion, when the authors have disclosed the results.

Answer: We exclude the sentence from Introduction.

Line 85-86:

No negative effects on adult storks were observed”.

I think this sentence is more appropriate to be included in the results rather than methods.

Answer: We exclude the sentence from Materials and Methods.  

Line 94-95:

A total of 36,050 records were collected from the movements of these individuals”.

I think this sentence is more appropriate to be included in the results rather than methods.

Answer: We move this sentence to Results (line 221)

In the results, the authors provide a number of average, but they do not give any standard deviation or range of the average

Answer: We have added a standard error in line 224. Table 1 shows the range of variation in the time spent by the black stork individuals at distance intervals

Comments on the Quality of English Language

I am not native English speaker, so I do not feel good enough to comment on this regard, but there are parts of the article that it is difficult to understand to me because of the English writing of this article. 

Answer: The manuscript has undergone English language editing by MDPI. The text has been checked for correct use of grammar and common technical terms, and edited to a level suitable for reporting research in a scholarly journal.

Reviewer 2 Report

Comments and Suggestions for Authors

See attached file

Comments on the Quality of English Language

Overall, the authors use high-quality English but frequently use the passive voice and inconsistently present names and numbers. 

Author Response

The authors explored young stork nest use and their willingness to use adjacent space. Overall the data is detailed and offers novel information but there is an need for more context and clarity in how the authors present the data.

38-italizize the genus name

Answer: Corrected line 41

40- italicize Latin name

Answer: Corrected line 44

43 spell out the EEU at first mention

Answer: Corrected line 47

43: add a citation

Answer: Corrected line 47

44: add a space between the numbers and the unit (200 m) -do this throughout the document

Answer: Corrected throughout the manuscript

45: what are the numbers after season?

Answer: That are date describing period of protection. We have rephrased the sentence to make it clearer  line 67.

40 and 43: black stork is incontinently presented. Most common names are lower case.

Answer: Corrected line 43-44.

48: same as above regarding black stork presentation

Answer: We have changed the name to BS throughout the manuscript

53: why are loggers and camera traps good tools foe these studies

Answer: We have rephrased the sentence to make it clearer  (Line 83-85)

71: 200- meter and 700-meter is presented in a new way compared to lines 44-45

Answer: We have rephrased the sentence to make it clearer (Line 65-67)

Why is it important to focus on the young storks. Include more about this in the introduction to justify the need for this study and the reason for focusing on this age class. Given the focus of the study and the information in the discussion the Introduction needs more background information of stork development and behavior in terms of leaving the nest and first migration.

Answer: Following the reviewer's suggestion, we have restructured the introduction and discussion. The goals of study have been reformulated to be more clear.

75: passive voice, update this sentence so it leads with the subject.

Answer: The manuscript has undergone English language editing.

What camera trap brand did you use?

Answer: We have provide the information about camera trap brand (line 116)

82: This sentence is also in the passive voice, please update

Answer: The manuscript has undergone English language editing.

86: what does MMS stand for?

Answer: Corrected line 126

89: The authors switch between photo logger and photo traps… use a consistent term for this tool or describe how they are distinct.

Answer: We use camera traps.

94: what brand?

Answer: Corrected. Line 116

108: passive voice

Answer: The manuscript has undergone English language editing.

12-123: passive voice

Answer: The manuscript has undergone English language editing.

165: italicize or use “” for R packages

Answer: Corrected line 215

What about the photo data?

Answer: Explanation in line 128

Figure: The scale should be smaller to show a better picture of what is happening between 0-200. It is unclear what decade of life means here since it is in days.

Answer: We have prepared new graph with logarithmic scale. Figure 2.

Were there any efforts to remove outliers?

Answer: Outliers for distances above 200 km have been omitted for better readability of the graph, but included in the statistical analysis.

Figure 4: the formatting is off on the y axis and in the legend. Can this be presented as a mean across all the storks?

Answer: We deleted this figure.

Discussion:

The discussion spent more time revisiting the study’s results instead of discussion how these results fit within the current literature. The conclusion offers clear insights about the data. But the discussion is lacking more context about young stork light behavior.

Answer: We have restructured the Discussion and we have added some literature positions.

Reviewer 3 Report

Comments and Suggestions for Authors

I have completed my review for 'Telemetry studies of space use by young black storks Ciconia nigra in the post-breeding period', which is currently under consideration for publication in Animals. This study describes young black storks' home range and behavior during the post-breeding stage in Poland. During the study, it was evident that a core area of 500 m around the nest was critical for young storks during the independence period. I enjoyed reading the manuscript and believe it could be essential to this species' breeding success and conservation. However, the manuscript has some issues that need further clarification.

The authors describe the photo dataset in the methods. However, it is not clear which results and discussions are derived from this dataset. Please, if possible, be more specific about how these data were used in the results and discussions.

In two years of study (2022 and 2023), 34 chicks were equipped with GPS-GSM loggers in two regions of Poland. However, it needs to be made clear how many of those were deployed in 2022 in the western part and how many in 2023 in central and southeastern Poland. More information about sample sizes, colony size, and overall breeding conditions is needed. Moreover, there were some differences in nesting phenology among locations, but year differences in breeding performance may confound these differences.

Interestingly, the number of chicks did not influence space use, and the broods did not behave similarly throughout the study period. Siblings may stay together during this stage to have similar resting places near the nest.

In the breeding phenology, it would be easier if the authors used the date rather than the day of the year.

The discussion (page 12. Lines 321-328) needs to be clarified where this information came from. No results were mentioned in the breeding phenology.  

Author Response

I have completed my review for 'Telemetry studies of space use by young black storks Ciconia nigra in the post-breeding period', which is currently under consideration for publication in Animals. This study describes young black storks' home range and behavior during the post-breeding stage in Poland. During the study, it was evident that a core area of 500 m around the nest was critical for young storks during the independence period. I enjoyed reading the manuscript and believe it could be essential to this species' breeding success and conservation. However, the manuscript has some issues that need further clarification.

The authors describe the photo dataset in the methods. However, it is not clear which results and discussions are derived from this dataset. Please, if possible, be more specific about how these data were used in the results and discussions.

 Answer: Explanation in line 128

In two years of study (2022 and 2023), 34 chicks were equipped with GPS-GSM loggers in two regions of Poland. However, it needs to be made clear how many of those were deployed in 2022 in the western part and how many in 2023 in central and southeastern Poland. More information about sample sizes, colony size, and overall breeding conditions is needed. Moreover, there were some differences in nesting phenology among locations, but year differences in breeding performance may confound these differences.

 Answer: We added the map of analysed nests. We added information about nesting trees also. 

Interestingly, the number of chicks did not influence space use, and the broods did not behave similarly throughout the study period. Siblings may stay together during this stage to have similar resting places near the nest.

 Answer: Interesting comment. Unfortunately, we had too few cases of sibling transmitters to tell.

In the breeding phenology, it would be easier if the authors used the date rather than the day of the year.

 Answer: We corrected it at the figure 5.

The discussion (page 12. Lines 321-328) needs to be clarified where this information came from. No results were mentioned in the breeding phenology. 

 Answer: We have reworked the passage to make it less speculative (line 376-384).